# LEARNABILITY FOR THE INFORMATION BOTTLENECK

**Tailin Wu**
MIT
tailin@mit.edu

**Ian Fischer**
Google AI
iansf@google.com

**Isaac L. Chuang**
MIT
ichuang@mit.edu

**Max Tegmark**
MIT
tegmark@mit.edu

## ABSTRACT

Compressed representations generalize better (Shamir et al., 2010), which may be crucial when learning from limited or noisy labeled data. The Information Bottleneck (IB) method (Tishby et al. (2000)) provides an insightful and principled approach for balancing compression and prediction in representation learning. The IB objective $I(X;Z) - \beta I(Y;Z)$ employs a Lagrange multiplier $\beta$ to tune this trade-off. However, there is little theoretical guidance for how to select $\beta$. There is also a lack of theoretical understanding about the relationship between $\beta$, the dataset, model capacity, and *learnability*. In this work, we show that if $\beta$ is improperly chosen, learning cannot happen: the trivial representation $P(Z|X) = P(Z)$ becomes the global minimum of the IB objective. We show how this can be avoided, by identifying a sharp phase transition between the unlearnable and the learnable which arises as $\beta$ varies. This phase transition defines the concept of *IB-Learnability*. We prove several sufficient conditions for IB-Learnability, providing theoretical guidance for selecting $\beta$. We further show that IB-learnability is determined by the largest *confident*, *typical*, and *imbalanced subset* of the training examples. We give a practical algorithm to estimate the minimum $\beta$ for a given dataset. We test our theoretical results on synthetic datasets, MNIST, and CIFAR10 with noisy labels, and make the surprising observation that accuracy may be non-monotonic in $\beta$.

## 1 INTRODUCTION AND RELATED WORK

Compressed representations generalize better (Shamir et al., 2010), which is likely to be particularly important when learning from limited or noisy labels, as otherwise we should expect our models to overfit to the noise. Tishby et al. (2000) introduced the *Information Bottleneck* (IB) objective function which learns a representation $Z$ of observed variables $(X, Y)$ that retains as little information about $X$ as possible, but simultaneously captures as much information about $Y$ as possible:

$$\min \mathrm{IB}_\beta(X, Y; Z) = \min I(X; Z) - \beta I(Y; Z) \tag{1}$$

$I(X;Y) = \int dx\, dy\, p(x,y)\log\frac{p(x,y)}{p(x)p(y)}$ is the mutual information. The hyperparameter $\beta$ controls the trade-off between compression and prediction, in the same spirit as Rate-Distortion Theory (Shannon, 1948), but with a learned representation function $P(Z|X)$ that automatically captures some part of the "semantically meaningful" information, where the semantics are determined by the observed relationship between $X$ and $Y$.

The IB framework has been extended to and extensively studied in a variety of scenarios, including Gaussian variables (Chechik et al. (2005)), meta-Gaussians (Rey & Roth (2012)), continuous variables via variational methods (Alemi et al. (2016); Chalk et al. (2016); Fischer (2018)), deterministic scenarios (Strouse & Schwab (2017a); Kolchinsky et al. (2019)), geometric clustering (Strouse & Schwab (2017b)), and is used for learning invariant and disentangled representations in deep neural nets (Achille & Soatto (2018a;b)). However, a core issue remains: how should we select $\beta$? In the original work, the authors recommend sweeping $\beta > 1$, which can be prohibitively expensive in practice, but also leaves open interesting theoretical questions around the relationship between $\beta$, $P(Z|X)$, and the observed data, $P(X, Y)$. For example, under how much label noise will IB at a given $\beta$ still be able to learn a useful representation?

This work begins to answer some of those questions by characterizing the *onset* of learning. Specifically:

- We show that improperly chosen $\beta$ may result in a failure to learn: the trivial solution $P(Z|X) = P(Z)$ becomes the global minimum of the IB objective, even for $\beta \gg 1$.

- We introduce the concept of *IB-Learnability*, and show that when we vary $\beta$, the IB objective will undergo a phase transition from the inability to learn to the ability to learn.

- Using the second-order variation, we derive sufficient conditions for IB-Learnability, which provide theoretical guidance for choosing a good $\beta$.

- We show that IB-learnability is determined by the largest *confident*, *typical*, and *imbalanced subset* of the training examples, reveal its relationship with the slope of the Pareto frontier at the origin on the *information plane* $I(Y; Z)$ vs. $I(X; Z)$, and discuss its relation with model capacity.

We use our main results to demonstrate on synthetic datasets, MNIST (LeCun et al., 1998), and CIFAR10 (Krizhevsky & Hinton, 2009) under noisy labels that the theoretical prediction for IB-Learnability closely matches experiment. We present an algorithm for estimating the onset of IB-Learnability, and demonstrate that it does a good job of estimating the theoretical predictions and the empirical results. Finally, we observe discontinuities in the Pareto frontier of the information plane as $\beta$ increases, and those dicontinuities correspond to accuracy *decreasing* as $\beta$ increases.

## 2 IB-LEARNABILITY AND ITS SUFFICIENT CONDITIONS

We are given instances of $(x, y) \in \mathcal{X} \times \mathcal{Y}$ drawn from a distribution with probability (density) $P(X, Y)$, where unless otherwise stated, both $X$ and $Y$ can be discrete or continuous variables. $(X, Y)$ is our *training data*, and may be characterized by different types of noise. We can learn a representation $Z$ of $X$ with conditional probability[1] $p(z|x)$, such that $X, Y, Z$ obey the Markov chain $Z \leftarrow X \leftrightarrow Y$. Eq. (1) above gives the IB objective with Lagrange multiplier $\beta$, $\text{IB}_\beta(X, Y; Z)$, which is a functional of $p(z|x)$: $\text{IB}_\beta(X, Y; Z) = \text{IB}_\beta[p(z|x)]$. The IB learning task is to find a conditional probability $p(z|x)$ that minimizes $\text{IB}_\beta(X, Y; Z)$. The larger $\beta$, the more the objective favors making a good prediction for $Y$. Conversely, the smaller $\beta$, the more the objective favors learning a *concise* representation.

How can we select $\beta$ such that the IB objective learns a useful representation? In practice, the selection of $\beta$ is done empirically. Indeed, Tishby et al. (2000) recommends "sweeping $\beta$". In this section, we provide theoretical guidance for choosing $\beta$ by introducing the concept of IB-Learnability and providing a series of IB-learnable conditions.

**Definition 1** (IB$_\beta$-Learnability). *$(X, Y)$ is IB$_\beta$-learnable if there exists a $Z$ given by some $p_1(z|x)$, such that $IB_\beta(X, Y; Z)|_{p_1(z|x)} < IB_\beta(X, Y; Z)|_{p(z|x)=p(z)}$, where $p(z|x) = p(z)$ characterizes the trivial representation such that $Z = Z_{trivial}$ is independent of $X$.*

If $(X; Y)$ is IB$_\beta$-learnable, then when $\text{IB}_\beta(X, Y; Z)$ is globally minimized, it will *not* learn a trivial representation. If $(X; Y)$ is not IB$_\beta$-learnable, then when $\text{IB}_\beta(X, Y; Z)$ is globally minimized, it may learn a trivial representation.

**Necessary condition for IB-Learnability.** From Definition 1, we can see that IB$_\beta$-Learnability for any dataset $(X; Y)$ requires $\beta > 1$. In fact, from the Markov chain $Z \leftarrow X \leftrightarrow Y$, we have $I(Y; Z) \le I(X; Z)$ via the data-processing inequality. If $\beta \le 1$, then since $I(X; Z) \ge 0$ and $I(Y; Z) \ge 0$, we have that $\min(I(X; Z) - \beta I(Y; Z)) = 0 = \text{IB}_\beta(X, Y; Z_{trivial})$. Hence $(X, Y)$ is not IB$_\beta$-learnable for $\beta \le 1$.

Theorem 1 characterizes the IB$_\beta$-Learnability range for $\beta$ (see Appendix B for the proof):

**Theorem 1.** *If $(X, Y)$ is IB$_{\beta_1}$-learnable, then for any $\beta_2 > \beta_1$, it is IB$_{\beta_2}$-learnable.*

Based on Theorem 1, the range of $\beta$ such that $(X, Y)$ is IB$_\beta$-learnable has the form $\beta \in (\beta_0, +\infty)$. Thus, $\beta_0$ is the *threshold* of IB-Learnability. Furthermore, the trivial representation is a stationary solution for the IB objective:

**Lemma 1.1.** *$p(z|x) = p(z)$ is a stationary solution for $IB_\beta(X, Y; Z)$.*

The proof in Appendix E shows that the first-order variation $\delta\text{IB}_\beta[p(z|x)] = 0$ vanishes at the trivial representation. Lemma 1.1 yields our strategy for finding sufficient conditions for learnability: find conditions such that $p(z|x) = p(z)$ is not a local minimum for the functional $\text{IB}_\beta[p(z|x)]$. By requiring that the second order variation $\delta^2\text{IB}_\beta[p(z|x)] < 0$ at the trivial representation (Suff. Cond. 1, Appendix C), and constructing a special form of perturbation at the trivial representation (Suff. Cond. 2, Appendix F), we arrive at the key result of this paper (see Appendix G for the proof)[2]:

**Theorem 2** (**Confident Subset Suff. Cond.**). *A sufficient condition for $(X, Y)$ to be IB$_\beta$-learnable is $X$ and $Y$ are not independent, and*

$$\beta > \inf_{\Omega_x \subset \mathcal{X}} \beta_0(\Omega_x) = \inf_{\Omega_x \subset \mathcal{X}} \frac{\frac{1}{p(\Omega_x)} - 1}{\mathbb{E}_{y \sim p(y|\Omega_x)} \left[ \frac{p(y|\Omega_x)}{p(y)} - 1 \right]} \tag{2}$$

*where $\Omega_x$ denotes the event that $x \in \Omega_x$, with probability $p(\Omega_x)$. Moreover, $(\inf_{\Omega_x \subset \mathcal{X}} \beta_0(\Omega_x))^{-1}$ gives a lower bound on the slope of the Pareto frontier at the origin of the information plane $I(Y; Z)$ vs. $I(X; Z)$.*

---

[1] We use capital letters $X, Y, Z$ for variables and lowercase $x, y, z$ to denote the instance of variables, with $P(\cdot)$ and $p(\cdot)$ denoting their probability or probability density, respectively.

[2] The theorems in this paper deal with learnability w.r.t. the true mutual information (MI). If parameterized models are used to approximate MI, the limitation of the model capacity will translate into more uncertainty about $Y$ given $X$, viewed from the model.

**Characteristics of dataset leading to low $\beta_0$.** From Eq. (2), we see that three characteristics of the subset $\Omega_x \subset \mathcal{X}$ lead to low $\beta_0$: **(1) confidence:** $p(y|\Omega_x)$ is large; **(2) typicality and size:** the number of elements in $\Omega_x$ is large, or the elements in $\Omega_x$ are typical, leading to a large probability of $p(\Omega_x)$; **(3) imbalance:** $p(y)$ is small for the subset $\Omega_x$, but large for its complement. In summary, $\beta_0$ will be determined by the largest *confident*, *typical* and *imbalanced subset* of examples, or an equilibrium of those characteristics.

Theorem 2 immediately leads to two important corollaries under special problem structures: classification with class-conditional noisy labels (Angluin & Laird (1988)) and deterministic mappings.

**Corollary 2.1.** *Suppose that the true class labels are $y^*$, and the input space belonging to each $y^*$ has no overlap. We only observe the corrupted labels $y$ with class-conditional noise $p(y|x, y^*) = p(y|y^*) \neq p(y)$. Then a sufficient condition for IB$_\beta$-Learnability is:*

$$\beta > \inf_{y^*} \frac{\frac{1}{p(y^*)} - 1}{\sum_y \frac{p(y|y^*)^2}{p(y)} - 1} \tag{3}$$

**Corollary 2.2.** *For classification problems, if $Y$ is a deterministic function of $X$ and not independent of $X$, then a necessary and sufficient condition for IB$_\beta$-Learnability is $\beta > \beta_0 = 1$.*

Therefore, if we find that $\beta_0 > 1$ for a classification task, we may infer that $Y$ is not a deterministic function of $X$, i.e. either some classes have overlap, or the labels are noisy. However, finite models may add effective class overlap if they have insufficient capacity for the learning task. This may translate into a higher observed $\beta_0$, even when learning deterministic functions. Proofs are provided in Appendix H.

# 3 ESTIMATING THE IB-LEARNABILITY CONDITION

Based on Theorem 2, for general classification tasks we suggest Algorithm 1 in Appendix J to empirically estimate an upper-bound $\tilde{\beta}_0 \geq \beta_0$. Here, we give the intuition behind the algorithm.

First, we train a single maximum likelihood model on the dataset. That model provides estimates for all $p(y|x)$ in the training set. Since learnability is defined with respect to the *training data*, it is correct to directly use the empirical probability of $p(x)$ and $p(y)$ in the training data. Given $p(x)$, $p(y)$, and $p(y|x)$, and the understanding that we are seaching for a confident subset $\Omega_x$, we can then perform an efficient targeted search of the exponential space of subsets of the training data. The algorithm returns the lowest estimate of $\tilde{\beta}_0$ found during that process.

After estimating $\tilde{\beta}_0$, we can then use it for learning with IB, either directly, or as an anchor for a region where we can perform a much smaller sweep than we otherwise would have. This may be particularly important for very noisy datasets, where $\beta_0$ can be very large.

# 4 EXPERIMENTS

To test our theoretical results and Alg. 1, we perform experiments on synthetic datasets, MNIST, and CIFAR10. Additional experiment details are provided in Appendix K.

**Synthetic datasets.** We generate a set of synthetic datasets with varying class-conditional noise rates. Fig. 1 shows the results of sweeping $\beta$ to find the empirical onset of learning, and compares that onset to the predicted onset using Eq. (3). Clearly the estimate provides a tight upper bound in this simple setting. Also note that $\beta_0$ grows exponentially as the label noise increases, underscoring that improperly-chosen $\beta$ may result in an inability to learn useful representations, and the importance of theoretically-guided $\beta$ selection as opposed to sweeping $\beta$ in general.

**MNIST.** We perform binary classification with digits 0 and 1, but again add class-conditional noise to the labels with varying noise rates $\rho$. To explore how the model capacity influences the onset of learning, for each dataset we train two sets of Variational Information Bottleneck (Alemi et al., 2016) (VIB) models differing only by the number of neurons in their hidden layers of the encoder: one with $n = 128$ neurons, the other with $n = 512$ neurons. Insufficient capacity will result in more uncertainty of $Y$ given $X$ from the point of view of the model, so we expect $\beta_{0,\text{observed}}$ for the $n = 128$ model to be larger. Fig. 1 confirms this prediction. It also shows the $\beta_{0,\text{estimated}}$ and $\beta_{0,\text{predicted}}$ given by Algorithm 1 and Eq. (3), respectively. We see that Algorithm 1 does a good job estimating the onset of learning for the large-capacity model, and that the estimated results line up well with the theoretical predictions.

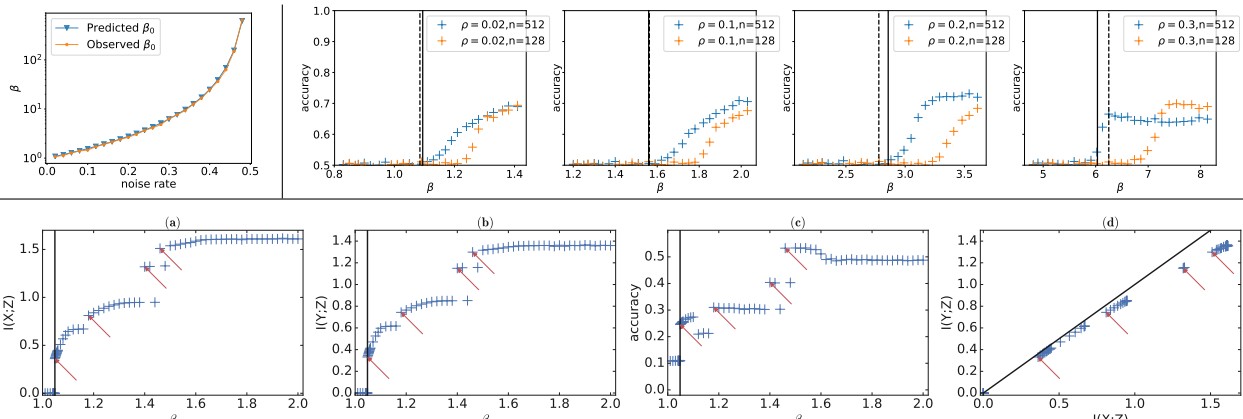

Figure 1: **Top Left:** Synthetic dataset predicted vs. experimentally identified $\beta_0$, with varying class-conditional noise. **Top Right:** MNIST dataset classification accuracy w.r.t. the true labels, for different noise rates $\rho$ and hidden units per layer $n$. The *dashed* vertical lines are $\beta_{0,\text{predicted}}$ by the R.H.S. of Eq. (3). The *solid* vertical lines are $\beta_{0,\text{estimated}}$ estimated using Alg. 1. $I(Y;Z)$ has identical behavior to accuracy, so we omit those results. $n = 128$ has insufficient capacity for the problem, so its learnability onset is pushed higher. At $\rho = 0.3$, $n = 512$ we can see that Eq. (3) is an upper bound on the true $\beta_0$. **Bottom:** CIFAR10 plots of $\beta$ vs **(a)** $I(X;Z)$, **(b)** $I(Y;Z)$, and **(c)** accuracy, as well as **(d)**, showing the information plane, all on the training set with 20% label noise. Each blue cross corresponds to a fully-converged model starting with independent initialization. The vertical black lines in **(a-c)** correspond to the predicted $\beta_0 = 1.0483$. The empirical $\hat{\beta}_0 = 1.048$. The diagonal black line in **(d)** is one of the hard boundaries of the information plane, $I(Y;Z) = I(X;Z)$. The Pareto frontier can never lie above that line. The red arrows indicate the models with lowest $\beta$ in the clearly distinct clusters in **(d)**. Accuracy **(c)** is non-monotonic in each of those clusters, especially the fourth, but $I(X;Z)$ **(a)** and $I(Y;Z)$ **(b)** are essentially monotonic.

**CIFAR10 forgetting.** For CIFAR10 (Krizhevsky & Hinton, 2009), we study how *forgetting* varies with $\beta$. In other words, given a VIB model trained at some high $\beta_2$, if we anneal it down to some much lower $\beta_1$, what accuracy does the model converge to? We estimated $\beta_0 = 1.0483$ on a version of CIFAR10 with 20% label noise using Alg. 1. The lowest $\beta$ with performance above chance was $\beta = 1.048$. See Appendix K.1 for experiment details.

As can be seen in Fig. 1 **(d)**, there are large discontinuities in the Pareto frontier, even though we vary $\beta$ in very small increments. Those discontinuities start at points on the Pareto frontier where many values of $\beta$ yield essentially the same $I(X;Z)$ and $I(Y;Z)$, and end when $\beta$ crosses apparent phase transitions that give large increases in *both* $I(X;Z)$ and $I(Y;Z)$ (marked with red arrows). Fig. 1 **(c)** shows that the lowest value of $\beta$ in each such region tends to have the highest accuracy.

A primary empirical result of our work is the following: some datasets have non-monotonic performance in regions where multiple values of $\beta$ cluster together. This surprising behavior is important to check for when training IB models. More thorough study is needed, but based on our initial results, we may expect that reducing $\beta$ to the minimal value that achieves a particular point on the information plane yields better representations. The phenomenon of discontinuities is also observed in prediction error vs. information in the model parameter (Achille & Soatto (2018a); Achille et al. (2019)), $I(c; X)$ vs. $H(c)$ ($c$ denotes clusters) in geometric clustering (Strouse & Schwab (2017b)). Although these discontinuities (including ours) are observed via different axes, we conjecture that they may all have a shared root cause, which is an interesting topic for future research.

## 5    CONCLUSION

In this paper, we have presented theoretical results for predicting the onset of learning, and have shown that it is determined by the largest confident, typical and imbalanced subset of the examples. We gave a practical algorithm for predicting the transition, and showed that those predictions are accurate, even in cases of extreme label noise. We have also observed a surprising non-monotonic relationship between $\beta$ and accuracy, and shown its relationship to discontinuities in the Pareto frontier of the information plane. We believe these results will provide theoretical and practical guidance for choosing $\beta$ in the IB framework for balancing prediction and compression. Our work also raises other questions, such as whether there are other phase transitions in learnability that might be identified. We hope to address some of those questions in future work.

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

# Appendix

The structure of the Appendix is as follows. In Appendix A, we provide preliminaries for the first-order and second-order variations on functionals. Then we prove Theorem 1 in Appendix B. In Appendix C, we state and prove Sufficient Condition 1 for $\text{IB}_\beta$-learnability. In Appendix D, we calculate the first and second variations of $\text{IB}_\beta[p(z|x)]$ at the trivial representation $p(z|x) = p(z)$, which is used in proving the Sufficient Condition 2 $\text{IB}_\beta$-learnability (Appendix F). After these preparations, we prove the key result of this paper, Theorem 2, in Appendix G. Then two important corollaries 2.1, 2.2 are proved in Appendix H. We provide additional discussions and insights for Theorem 2 in Appendix I, and Algorithm 1 for estimation of an upper bound $\tilde{\beta}_0 \geq \beta_0$ in Appendix J. Finally in Appendix K, we provide details for the experiments.

## A  PRELIMINARIES: FIRST-ORDER AND SECOND-ORDER VARIATION

Let functional $F[f(x)]$ be defined on some normed linear space $\mathscr{R}$. Let us add a perturbative function $\epsilon h(x)$ to $f(x)$, and now the functional $F[f(x) + \epsilon h(x)]$ can be expanded as

$$\Delta F[f(x)] = F[f(x) + \epsilon h(x)] - F[f(x)]$$
$$= \varphi_1[f(x)] + \varphi_2[f(x)] + \mathcal{O}(\epsilon^3 ||h||^2)$$

where $||h||$ denotes the norm of $h$, $\varphi_1[f(x)] = \epsilon \frac{dF[f(x)]}{d\epsilon}$ is a linear functional of $\epsilon h(x)$, and is called the *first-order variation*, denoted as $\delta F[f(x)]$. $\varphi_2[f(x)] = \frac{1}{2}\epsilon^2 \frac{d^2 F[f(x)]}{d\epsilon^2}$ is a quadratic functional of $\epsilon h(x)$, and is called the *second-order variation*, denoted as $\delta^2 F[f(x)]$.

If $\delta F[f(x)] = 0$, we call $f(x)$ a stationary solution for the functional $F[\cdot]$.

If $\Delta F[f(x)] \geq 0$ for all $h(x)$ such that $f(x) + \epsilon h(x)$ is at the neighborhood of $f(x)$, we call $f(x)$ a (local) minimum of $F[\cdot]$.

## B  PROOF OF THEOREM 1

*Proof.* At the trivial representation $p(z|x) = p(z)$, we have $I(X;Z) = 0$, and $I(Y;Z) = 0$ due to the Markov chain, so $\text{IB}_\beta(X,Y;Z)|_{p(z|x)=p(z)} = 0$ for any $\beta$. Since $(X,Y)$ is $\text{IB}_{\beta_1}$-learnable, there exists a $Z$ given by a $p_1(z|x)$ such that $\text{IB}_{\beta_1}(X,Y;Z)|_{p_1(z|x)} < 0$. Since $\beta_2 > \beta_1$, and $I(Y;Z) \geq 0$, we have $\text{IB}_{\beta_2}(X,Y;Z)|_{p_1(z|x)} \leq \text{IB}_{\beta_1}(X,Y;Z)|_{p_1(z|x)} < 0 = \text{IB}_{\beta_2}(X,Y;Z)|_{p(z|x)=p(z)}$. Therefore, $(X,Y)$ is $\text{IB}_{\beta_2}$-learnable. $\square$

## C  SUFFICIENT CONDITION 1 AND PROOF

In this section, we prove the Sufficient Condition 1 for $\text{IB}_\beta$-learnability, which will lay the foundation for the Sufficient condition 2 (Appendix F) and the Confident Subset Sufficient condition (key result of this paper, Theorem 2) that follow.

**Theorem 3 (Suff. Cond. 1).** *A sufficient condition for $(X,Y)$ to be $\text{IB}_\beta$-learnable is that there exists a perturbation function $h(z|x)$ with[3] $\int h(z|x)dz = 0$, such that the second-order variation $\delta^2 IB_\beta[p(z|x)] < 0$ at the trivial representation $p(z|x) = p(z)$.*

*Proof.* To prove Theorem 3, we use the Theorem 1 of Chapter 5 of Gelfand et al. (2000) which gives a necessary condition for $F[f(x)]$ to have a minimum at $f_0(x)$. Adapting to our notation, we have:

**Theorem 4** (Gelfand et al. (2000)). *A necessary condition for the functional $F[f(x)]$ to have a minimum at $f(x) = f_0(x)$ is that for $f(x) = f_0(x)$ and all admissible $\epsilon h(x)$,*

$$\delta^2 F[f(x)] \geq 0$$

.

---

[3]Whenever a variable $W$ is discrete, we can simply replace the integral $(\int \cdot dw)$ by summation $(\sum_w \cdot)$.

Applying to our functional $IB_\beta[p(z|x)]$, an immediate result of Theorem 4 is that, if at $p(z|x) = p(z)$, there exists an $\epsilon h(z|x)$ such that $\delta^2 IB_\beta[p(z|x)] < 0$, then $p(z|x) = p(z)$ is not a minimum for $IB_\beta[p(z|x)]$. Using the definition of $IB_\beta$ learnability, we have that $(X, Y)$ is $IB_\beta$-learnable.

$\square$

Intuitively, if $\delta^2 IB_\beta[p(z|x)]\big|_{p(z|x)=p(z)} < 0$, we can always find a $p'(z|x) = p(z|x) + h(z|x)$ in the neighborhood of the trivial representation $p(z|x) = p(z)$, such that $IB_\beta[p'(z|x)] < IB_\beta[p(z|x)]$, thus satisfying the definition for $IB_\beta$-Learnability.

To make Theorem 3 more practical, we perturb $p(z|x)$ around the trivial solution $p'(z|x) = p(z|x) + \epsilon h(z|x)$, and expand $IB_\beta[p(z|x) + h(z|x)] - IB_\beta[p(z|x)]$ to the second order of $\epsilon$. We can then prove Theorem 5:

## D    FIRST- AND SECOND-ORDER VARIATIONS OF $IB_\beta[p(z|x)]$

In this section, we derive the first- and second-order variations of $IB_\beta[p(z|x)]$, which are needed for proving Lemma 1.1 and Theorem 5.

**Lemma 4.1.** *Using perturbative function $h(z|x)$, we have*

$$\delta IB_\beta[p(z|x)] = \int dxdzp(x)h(z|x)log\frac{p(z|x)}{p(z)} - \beta \int dxdydzp(x,y)h(z|x)log\frac{p(z|y)}{p(z)}$$

$$\delta^2 IB_\beta[p(z|x)] =$$
$$\frac{1}{2}\left[ \int dxdz\frac{p(x)^2}{p(x,z)}h(z|x)^2 - \beta \int dxdx'dydz\frac{p(x,y)p(x',y)}{p(y,z)}h(z|x)h(z|x') + (\beta-1)\int dxdx'dz\frac{p(x)p(x')}{p(z)}h(z|x)h(z|x') \right]$$

*Proof.* Since $IB_\beta[p(z|x)] = I(X;Z) - \beta I(Y;Z)$, let us calculate the first and second-order variation of $I(X;Z)$ and $I(Y;Z)$ w.r.t. $p(z|x)$, respectively. Through this derivation, we use $\epsilon h(z|x)$ as a perturbative function, for ease of deciding different orders of variations. We will finally absorb $\epsilon$ into $h(z|x)$.

Denote $I(X;Z) = F_1[p(z|x)]$. We have

$$F_1[p(z|x)] = I(X;Z) = \int dxdzp(z|x)p(x)log\frac{p(z|x)}{p(z)}$$

Since

$$p(z) = \int p(z|x)p(x)dx$$

We have

$$p(z)|_{p(z|x)+\epsilon h(z|x)} = p(z)|_{p(z|x)} + \epsilon \int h(z|x)p(x)dx$$

Expanding $F_1[p(z|x) + \epsilon h(z|x)]$ to the second order of $\epsilon$, we have

$$F_1[p(z|x) + \epsilon h(z|x)]$$

$$= \int dx dz p(x)[p(z|x) + \epsilon h(z|x)]\log\frac{p(z|x) + \epsilon h(z|x)}{p(z) + \epsilon \int h(z|x')p(x')dx'}$$

$$= \int dx dz p(x)p(z|x)\left(1 + \epsilon\frac{h(z|x)}{p(z|x)}\right)\log\frac{p(z|x)\left(1 + \epsilon\frac{h(z|x)}{p(z|x)}\right)}{p(z)\left(1 + \epsilon\frac{\int h(z|x')p(x')dx'}{p(z)}\right)}$$

$$= \int dx dz p(x)p(z|x)\left(1 + \epsilon\frac{h(z|x)}{p(z|x)}\right)\log\left[\frac{p(z|x)}{p(z)}\left(1 + \epsilon\frac{h(z|x)}{p(z|x)}\right)\left(1 - \epsilon\frac{\int h(z|x')p(x')dx'}{p(z)}\right.\right.$$

$$+ \epsilon^2\left(\frac{\int h(z|x')p(x')dx'}{p(z)}\right)^2\left.\left.\right)\right] + \mathcal{O}(\epsilon^3)$$

$$= \int dx dz p(x)p(z|x)\left(1 + \epsilon\frac{h(z|x)}{p(z|x)}\right)\log\left[\frac{p(z|x)}{p(z)}\left(1 + \epsilon\left(\frac{h(z|x)}{p(z|x)} - \frac{\int h(z|x')p(x')dx'}{p(z)}\right)\right.\right.$$

$$+ \epsilon^2\left(\frac{\int h(z|x')p(x')dx'}{p(z)}\right)^2 - \epsilon^2\frac{h(z|x)}{p(z|x)}\frac{\int h(z|x')p(x')dx'}{p(z)}\left.\left.\right)\right] + \mathcal{O}(\epsilon^3)$$

$$= \int dx dz p(x)p(z|x)\left(1 + \epsilon\frac{h(z|x)}{p(z|x)}\right)\left[\log\frac{p(z|x)}{p(z)} + \epsilon\left(\frac{h(z|x)}{p(z|x)} - \frac{\int h(z|x')p(x')dx'}{p(z)}\right)\right.$$

$$+ \epsilon^2\left(\frac{\int h(z|x')p(x')dx'}{p(z)}\right)^2 - \epsilon^2\frac{h(z|x)}{p(z|x)}\frac{\int h(z|x')p(x')dx'}{p(z)} - \frac{1}{2}\epsilon^2\left(\frac{h(z|x)}{p(z|x)} - \frac{\int h(z|x')p(x')dx'}{p(z)}\right)^2\left.\right] + \mathcal{O}(\epsilon^3)$$

Collecting the first order terms of $\epsilon$, we have

$$\delta F_1[p(z|x)]$$

$$= \epsilon\int dx dz p(x)p(z|x)\left(\frac{h(z|x)}{p(z|x)} - \frac{\int h(z|x')p(x')dx'}{p(z)}\right) + \epsilon\int dx dz p(x)p(z|x)\frac{h(z|x)}{p(z|x)}\log\frac{p(z|x)}{p(z)}$$

$$= \epsilon\int dx dz p(x)h(z|x) - \epsilon\int dx' dz p(x')h(z|x') + \epsilon\int dx dz p(x)h(z|x)\log\frac{p(z|x)}{p(z)}$$

$$= \epsilon\int dx dz p(x)h(z|x)\log\frac{p(z|x)}{p(z)}$$

Collecting the second order terms of $\epsilon^2$, we have

$$\delta^2 F_1[p(z|x)]$$

$$= \epsilon^2\int dx dz p(x)p(z|x)\left[\left(\frac{\int h(z|x')p(x')dx'}{p(z)}\right)^2 - \frac{h(z|x)}{p(z|x)}\frac{\int h(z|x')p(x')dx'}{p(z)} - \frac{1}{2}\left(\frac{h(z|x)}{p(z|x)} - \frac{\int h(z|x')p(x')dx'}{p(z)}\right)^2\right]$$

$$+ \epsilon^2\int dx dz p(x)p(z|x)\frac{h(z|x)}{p(z|x)}\left(\frac{h(z|x)}{p(z|x)} - \frac{\int h(z|x')p(x')dx'}{p(z)}\right)$$

$$= \frac{\epsilon^2}{2}\int dx dz\frac{p(x)^2}{p(x,z)}h(z|x)^2 - \frac{\epsilon^2}{2}\int dx dx' dz\frac{p(x)p(x')}{p(z)}h(z|x)h(z|x')$$

Now let us calculate the first and second-order variation of $F_2[p(z|x)] = I(Z; Y)$. We have

$$F_2[p(z|x)] = I(Y; Z) = \int dy dz p(z|y)p(y)\log\frac{p(y,z)}{p(y)p(z)} = \int dx dy dz p(z|y)p(x,y)\log\frac{p(y,z)}{p(y)p(z)}$$

Using the Markov chain $Z \leftarrow X \leftrightarrow Y$, we have

$$p(y,z) = \int p(z|x)p(x,y)dx$$

Hence

$$p(y,z)|_{p(z|x)+\epsilon h(z|x)} = p(y,z)|_{p(z|x)} + \epsilon \int h(z|x)p(x,y)dx$$

Then expanding $F_2[p(z|x) + \epsilon h(z|x)]$ to the second order of $\epsilon$, we have

$$F_2[p(z|x) + \epsilon h(z|x)]$$

$$= \int dxdydzp(x,y)p(z|x)\left(1 + \epsilon\frac{h(z|x)}{p(z|x)}\right)\log\frac{p(y,z)\left(1+\epsilon\frac{\int h(z|x')p(x',y)dx'}{p(y,z)}\right)}{p(y)p(z)(1+\epsilon\frac{\int h(z|x'')p(x'')dx''}{p(z)})}$$

$$= \int dxdydzp(x,y)p(z|x)\left(1 + \epsilon\frac{h(z|x)}{p(z|x)}\right)\left[\log\frac{p(y,z)}{p(y)p(z)} + \epsilon\left(\frac{\int h(z|x')p(x',y)dx'}{p(y,z)} - \frac{\int h(z|x')p(x')dx'}{p(z)}\right)\right.$$

$$+ \epsilon^2\left[\left(\frac{\int h(z|x')p(x')dx'}{p(z)}\right)^2 - \frac{\int h(z|x')p(x',y)dx'}{p(y,z)}\frac{\int h(z|x'')p(x'')dx''}{p(z)} - \frac{1}{2}\left(\frac{\int h(z|x')p(x',y)dx'}{p(y,z)} - \frac{\int h(z|x')p(x')dx'}{p(z)}\right)^2\right]$$

$$+ \mathcal{O}(\epsilon^3)$$

Collecting the first order terms of $\epsilon$, we have

$$\delta F_2[p(z|x)]$$

$$= \epsilon\int dxdydzp(x,y)h(z|x)\log\frac{p(y,z)}{p(y)p(z)} + \epsilon\int dxdydzp(x,y)p(z|x)\frac{\int h(z|x')p(x',y)dx'}{p(y,z)}$$

$$- \epsilon\int dxdydzp(x,y)p(z|x)\frac{\int h(z|x')p(x')dx'}{p(z)}$$

$$= \epsilon\int dxdydzp(x,y)h(z|x)\log\frac{p(y,z)}{p(y)p(z)} + \epsilon\int dx'dydzh(z|x')p(x',y) - \epsilon\int dzh(z|x')p(x')dx'$$

$$= \epsilon\int dxdydzp(x,y)h(z|x)\log\frac{p(z|y)}{p(z)}$$

Collecting the second order terms of $\epsilon$, we have

$$\delta^2 F_2[p(z|x)]$$

$$= \epsilon^2\int dxdydzp(x,y)p(z|x)\left[\left(\frac{\int h(z|x')p(x')dx'}{p(z)}\right)^2 - \frac{\int h(z|x')p(x',y)dx'}{p(y,z)}\frac{\int h(z|x'')p(x'')dx''}{p(z)}\right]$$

$$- \frac{\epsilon^2}{2}\int dxdydzp(x,y)p(z|x)\left(\frac{\int h(z|x')p(x',y)dx'}{p(y,z)} - \frac{\int h(z|x')p(x')dx'}{p(z)}\right)^2$$

$$+ \epsilon^2\int dxdydzp(x,y)p(z|x)\frac{h(z|x)}{p(z|x)}\left(\frac{\int h(z|x')p(x',y)dx'}{p(y,z)} - \frac{\int h(z|x')p(x')dx'}{p(z)}\right)$$

$$= \frac{\epsilon^2}{2}\int dxdx'dydz\frac{p(x,y)p(x',y)}{p(y,z)}h(z|x)h(z|x') - \frac{\epsilon^2}{2}\int dxdx'dz\frac{p(x)p(x')}{p(z)}h(z|x)h(z|x')$$

Finally, we have

$$\delta\text{IB}_\beta[p(z|x)] = \delta F_1[p(z|x)] - \beta\cdot\delta F_2[p(z|x)]$$

$$= \epsilon\left(\int dxdzp(x)h(z|x)\log\frac{p(z|x)}{p(z)} - \beta\int dxdydzp(x,y)h(z|x)\log\frac{p(z|y)}{p(z)}\right) \quad (4)$$

$$
\begin{aligned}
\delta^2 \mathrm{IB}_\beta[p(z|x)] =& \delta^2 F_1[p(z|x)] - \beta \cdot \delta^2 F_2[p(z|x)] \\
=& \frac{\epsilon^2}{2} \int dx dz \frac{p(x)^2}{p(x,z)} h(z|x)^2 - \frac{\epsilon^2}{2} \int dx dx' dz \frac{p(x)p(x')}{p(z)} h(z|x)h(z|x') \\
& - \beta\epsilon^2 \left[ \frac{1}{2} \int dx dx' dy dz \frac{p(x,y)p(x',y)}{p(y,z)} h(z|x)h(z|x') - \frac{1}{2} \int dx dx' dz \frac{p(x)p(x')}{p(z)} h(z|x)h(z|x') \right] \\
=& \frac{\epsilon^2}{2} \left[ \int dx dz \frac{p(x)^2}{p(x,z)} h(z|x)^2 \right. \\
& \left. - \beta \int dx dx' dy dz \frac{p(x,y)p(x',y)}{p(y,z)} h(z|x)h(z|x') + (\beta-1) \int dx dx' dz \frac{p(x)p(x')}{p(z)} h(z|x)h(z|x') \right]
\end{aligned}
$$

Absorb $\epsilon$ into $h(z|x)$, we get rid of the $\epsilon$ factor and obtain the final expression in Lemma 4.1.

$\square$

## E  PROOF OF LEMMA 1.1

*Proof.* Using Lemma 4.1, we have

$$
\delta \mathrm{IB}_\beta[p(z|x)] = \int dx dz p(x) h(z|x) \log \frac{p(z|x)}{p(z)} - \beta \int dx dy dz p(x,y) h(z|x) \log \frac{p(z|y)}{p(z)}
$$

Let $p(z|x) = p(z)$ (the trivial representation), we have that $\log \frac{p(z|x)}{p(z)} \equiv 0$. Therefore, the two integrals are both 0. Hence,

$$
\delta \mathrm{IB}_\beta[p(z|x)] \big|_{p(z|x)=p(z)} \equiv 0
$$

Therefore, the $p(z|x) = p(z)$ is a stationary solution for $\mathrm{IB}_\beta[p(z|x)]$.

$\square$

## F  SUFFICIENT CONDITION 2 AND PROOF

### F.1  STATEMENT OF THE THEOREM

**Theorem 5 (Suff. Cond. 2).** *A sufficient condition for $(X, Y)$ to be $\mathrm{IB}_\beta$-learnable is $X$ and $Y$ are not independent, and*

$$
\beta > \inf_{h(x)} \beta_0[h(x)] \tag{5}
$$

*where the functional $\beta_0[h(x)]$ is given by*

$$
\beta_0[h(x)] = \frac{\frac{\mathbb{E}_{x \sim p(x)}[h(x)^2]}{\left(\mathbb{E}_{x \sim p(x)}[h(x)]\right)^2} - 1}{\mathbb{E}_{y \sim p(y)} \left[ \left( \frac{\mathbb{E}_{x \sim p(x|y)}[h(x)]}{\mathbb{E}_{x \sim p(x)}[h(x)]} \right)^2 \right] - 1}
$$

*Moreover, we have that $\left( \inf_{h(x)} \beta[h(x)] \right)^{-1}$ is a lower bound of the slope of the Pareto frontier in the information plane $I(Y; Z)$ vs. $I(X; Z)$ at the origin.*

The proof is given in Appendix F, which also gives a construction for $h(z|x)$ for Theorem 3 for any $h(x)$ satisfying Theorem 5, and shows that the converse is also true: if there exists $h(z|x)$ suth that the condition in Theorem 3 is true, then we can find $h(x)$ satisfying the the condition in Theorem 5.

The geometric meaning of $(\beta_0[h(x)])^{-1}$ is as follows. It equals $\frac{\Delta I(Y;Z)}{\Delta I(X;Z)} \big|_{p(z|x)=p(z)}$ under a perturbation function of the form $h_1(z|x) = h(x)h_2(z)$ (satisfying $\int h_2(z)dz = 0$ and $\int \frac{h_2^2(z)}{p(z)}dz > 0$) at the trivial representation, where $\Delta I(X; Z) = I(X; Z)|_{p(z|x)+h_1(z|x)} - I(X; Z)|_{p(z|x)}$ and similarly for $I(Y; Z)$. Since the first order

variation vanishes for both $I(X;Z)$ and $I(Y;Z)$, we have $\frac{\Delta I(Y;Z)}{\Delta I(X;Z)} = \frac{\delta^2 I(Y;Z)}{\delta^2 I(X;Z)}$, which turns out to be equal to $(\beta_0[h(x)])^{-1}$. Therefore, $\left(\inf_{h(x)} \beta[h(x)]\right)^{-1}$ gives the highest $\frac{\Delta I(Y;Z)}{\Delta I(X;Z)}\big|_{p(z|x)=p(z)}$ under the class of perturbation functions $h_1(z|x) = h_2(z)h(x)$, and provides a lower bound of $\sup_{h(z|x)} \frac{\Delta I(Y;Z)}{\Delta I(X;Z)}\big|_{p(z|x)=p(z)}$, which is the slope of the Pareto frontier in the information plane $I(Y;Z)$ vs. $I(X;Z)$ at the origin. Theorem 5 in essence states that as long as $\beta^{-1}$ is lower than this lower bound of the slope of the Pareto frontier, $(X;Y)$ is IB$_\beta$-learnable.

From Theorem 5, we see that it still has an infimum over an arbitrary function $h(x)$, which is not easy to estimate. To get rid of $h(x)$, we can use a specific functional form for $h(x)$ in Eq. (5), and obtain a stronger sufficient condition for IB$_\beta$-Learnability. But we want to choose $h(x)$ as near to the infimum as possible. To do this, we note the following characteristics for the R.H.S of Eq. (5):

- We can set $h(x)$ to be nonzero if $x \in \Omega_x$ for some region $\Omega_x \subset \mathcal{X}$ and 0 otherwise. Then we obtain the following sufficient condition:

$$\beta > \inf_{h(x), \Omega_x \in \mathcal{X}} \frac{\frac{\mathbb{E}_{x \sim p(x), x \in \Omega_x}[h(x)^2]}{\left(\mathbb{E}_{x \sim p(x), x \in \Omega_x}[h(x)]\right)^2} - 1}{\int \frac{dy}{p(y)} \left(\frac{\mathbb{E}_{x \sim p(x), x \in \Omega_x}[p(y|x)h(x)]}{\mathbb{E}_{x \sim p(x), x \in \Omega_x}[h(x)]}\right)^2 - 1} \tag{6}$$

- The numerator of the R.H.S. of Eq. (6) attains its minimum when $h(x)$ is a constant within $\Omega_x$. This can be proved using the Cauchy-Schwarz inequality: $\langle u, u \rangle \langle v, v \rangle \geq \langle u, v \rangle^2$, setting $u(x) = h(x)\sqrt{p(x)}$, $v(x) = \sqrt{p(x)}$, and defining the inner product as $\langle u, v \rangle = \int u(x)v(x)dx$. Therefore, the numerator of the R.H.S. of Eq. (6) $\geq \frac{1}{\int_{x \in \Omega_x} p(x)} - 1$, and attains equality when $\frac{u(x)}{v(x)} = h(x)$ is constant.

Based on these observations, we can let $h(x)$ be a nonzero constant inside some region $\Omega_x \subset \mathcal{X}$ and 0 otherwise, and the infimum over an arbitrary function $h(x)$ is simplified to infimum over $\Omega_x \subset \mathcal{X}$, and we obtain the *confident subset sufficient condition* (Theorem 2) for IB$_\beta$-Learnability, which is a key result of this paper.

### F.2 PROOF OF THEOREM 5 (SUFF. COND. 2)

*Proof.* Firstly, from the necessary condition of $\beta > 1$ in Section 2, we have that any sufficient condition for IB$_\beta$-learnability should be able to deduce $\beta > 1$.

Now using Theorem 3, a sufficient condition for $(X, Y)$ to be IB$_\beta$-learnable is that there exists $h(z|x)$ with $\int h(z|x)dx = 0$ such that $\delta^2 IB_\beta[p(z|x)] < 0$ at $p(z|x) = p(x)$.

At the trivial representation, $p(z|x) = p(z)$ and hence $p(x, z) = p(x)p(z)$. Due to the Markov chain $Z \leftarrow X \leftrightarrow Y$, we have $p(y, z) = p(y)p(z)$. Substituting them into the $\delta^2 IB_\beta[p(z|x)]$ in Lemma 4.1, the condition becomes: there exists $h(z|x)$ with $\int h(z|x)dz = 0$, such that

$$0 > \delta^2 IB_\beta[p(z|x)] =$$
$$\frac{1}{2}\left[\int dxdz \frac{p(x)^2}{p(x)p(z)}h(z|x)^2 - \beta \int dxdx'dydz \frac{p(x,y)p(x',y)}{p(y)p(z)}h(z|x)h(z|x') + (\beta-1)\int dxdx'dz \frac{p(x)p(x')}{p(z)}h(z|x)h(z|x')\right] \tag{7}$$

Rearranging terms and simplifying, we have

$$\int \frac{dz}{p(z)} G[h(z|x)] = \int \frac{dz}{p(z)}\left[\int dxh(z|x)^2 p(x) - \beta \int \frac{dy}{p(y)}\left(\int dxh(z|x)p(x)p(y|x)\right)^2 + (\beta-1)\left(\int dxh(z|x)p(x)\right)^2\right] < 0$$

where

$$G[h(x)] = \int dxh(x)^2 p(x) - \beta \int \frac{dy}{p(y)}\left(\int dxh(x)p(x)p(y|x)\right)^2 + (\beta-1)\left(\int dxh(x)p(x)\right)^2$$

Now we prove that the condition that $\exists h(z|x)$ s.t. $\int \frac{dz}{p(z)} G[h(z|x)] < 0$ is equivalent to the condition that $\exists h(x)$ s.t. $G[h(x)] < 0$.

If $\forall h(z|x)$, $G[h(z|x)] \geq 0$, then we have $\forall h(z|x)$, $\int \frac{dz}{p(z)} G[h(z|x)] \geq 0$. Therefore, if $\exists h(z|x)$ s.t. $\int \frac{dz}{p(z)} G[h(z|x)] < 0$, we have that $\exists h(z|x)$ s.t. $G[h(z|x)] < 0$. Since the functional $G[h(z|x)]$ does not contain integration over $z$, we can treat the $z$ in $G[h(z|x)]$ as a parameter and we have that $\exists h(x)$ s.t. $G[h(x)] < 0$.

Conversely, if there exists an certain function $h(x)$ such that $G[h(x)] < 0$, we can find some $h_2(z)$ such that $\int h_2(z)dz = 0$ and $\int \frac{h_2^2(z)}{p(z)} dz > 0$, and let $h_1(z|x) = h(x)h_2(z)$. Now we have

$$\int \frac{dz}{p(z)} G[h(z|x)] = \int \frac{h_2^2(z)dz}{p(z)} G[h(x)] = G[h(x)] \int \frac{h_2^2(z)dz}{p(z)} < 0$$

In other words, the condition Eq. (7) is equivalent to requiring that there exists an $h(x)$ such that $G[h(x)] < 0$. Hence, a sufficient condition for IB$_\beta$-learnability is that there exists an $h(x)$ such that

$$G[h(x)] = \int dx h(x)^2 p(x) - \beta \int \frac{dy}{p(y)} \left( \int dx h(x)p(x)p(y|x) \right)^2 + (\beta - 1) \left( \int dx h(x)p(x) \right)^2 < 0 \quad (8)$$

When $h(x) = C = $ const in the entire input space $\mathcal{X}$, Eq. (8) becomes:

$$C^2 - \beta C^2 + (\beta - 1)C^2 < 0$$

which cannot be true. Therefore, $h(x) = $ const cannot satisfy Eq. (8).

Rearranging terms and simplifying, and note that $\left[ \int dx h(x)p(x) \right]^2 > 0$ due to $h(x) \not\equiv 0 = $ const, we have

$$\beta \left[ \frac{\int \frac{dy}{p(y)} \left( \int dx h(x)p(x)p(y|x) \right)^2}{\left( \int dx h(x)p(x) \right)^2} - 1 \right] > \frac{\int dx h(x)^2 p(x)}{\left( \int dx h(x)p(x) \right)^2} - 1 \quad (9)$$

For the R.H.S. of Eq. (9), let us show that it is greater than 0. Using Cauchy-Schwarz inequality: $\langle u, u \rangle \langle v, v \rangle \geq \langle u, v \rangle^2$, and setting $u(x) = h(x)\sqrt{p(x)}$, $v(x) = \sqrt{p(x)}$, and defining the inner product as $\langle u, v \rangle = \int u(x)v(x)dx$. We have

$$\frac{\int dx h(x)^2 p(x)}{\left( \int dx h(x)p(x) \right)^2} \geq \frac{1}{\int p(x)dx} = 1$$

It attains equality when $\frac{u(x)}{v(x)} = h(x)$ is constant. Since $h(x)$ cannot be constant, we have that the R.H.S. of Eq. (9) is greater than 0.

For the L.H.S. of Eq. (9), due to the necessary condition that $\beta > 0$, if $\left[ \frac{\int \frac{dy}{p(y)} \left( \int dx h(x)p(x)p(y|x) \right)^2}{\left( \int dx h(x)p(x) \right)^2} - 1 \right] \leq 0$, Eq. (9) cannot hold. Then the $h(x)$ such that Eq. (9) holds is for those that satisfies

$$\frac{\int \frac{dy}{p(y)} \left( \int dx h(x)p(x)p(y|x) \right)^2}{\left( \int dx h(x)p(x) \right)^2} - 1 > 0$$

i.e.

$$\int dy p(y) \left( \int dx p(x|y)h(x) \right)^2 > \left( \int dx p(x)h(x) \right)^2$$

We see this constraint contains the requirement that $h(x) \not\equiv $ const.

Written in the form of expectations, we have

$$\mathbb{E}_{y \sim p(y)} \left[ \left( \mathbb{E}_{x \sim p(x|y)}[h(x)] \right)^2 \right] > \left( \mathbb{E}_{x \sim p(x)}[h(x)] \right)^2 \quad (10)$$

Since the square function is convex, using Jensen's inequality on the outer expectation on the L.H.S. of Eq. (10), we have

$$\mathbb{E}_{y\sim p(y)}\left[\left(\mathbb{E}_{x\sim p(x|y)}[h(x)]\right)^2\right] \geq \left(\mathbb{E}_{y\sim p(y)}\left[\mathbb{E}_{x\sim p(x|y)}[h(x)]\right]\right)^2 = \left(\mathbb{E}_{x\sim p(x)}[h(x)]\right)^2$$

The equality holds iff $\mathbb{E}_{x\sim p(x|y)}[h(x)]$ is constant w.r.t. $y$, i.e. $Y$ is independent of $X$. Therefore, in order for Eq. (10) to hold, we require that $Y$ is not independent of $X$.

Using Jensen's inequality on the innter expectation on the L.H.S. of Eq. (10), we have

$$\mathbb{E}_{y\sim p(y)}\left[\left(\mathbb{E}_{x\sim p(x|y)}[h(x)]\right)^2\right] \leq \mathbb{E}_{y\sim p(y)}\left[\mathbb{E}_{x\sim p(x|y)}[h(x)^2]\right] = \mathbb{E}_{x\sim p(x)}[h(x)^2] \tag{11}$$

The equality holds when $h(x)$ is a constant. Since we require that $h(x)$ is not a constant, we have that the equality cannot be reached.

Under the constraint that $Y$ is not independent of $X$, we can divide both sides of Eq. 8, and obtain the condition: there exists an $h(x)$ such that

$$\beta > \frac{\frac{\int dx h(x)^2 p(x)}{\left(\int dx h(x) p(x)\right)^2} - 1}{\int \frac{dy}{p(y)}\frac{\left(\int dx h(x) p(x) p(y|x)\right)^2}{\left(\int dx h(x) p(x)\right)^2} - 1}$$

i.e.

$$\beta > \inf_{h(x)} \frac{\frac{\int dx h(x)^2 p(x)}{\left(\int dx h(x) p(x)\right)^2} - 1}{\int \frac{dy}{p(y)}\frac{\left(\int dx h(x) p(x) p(y|x)\right)^2}{\left(\int dx h(x) p(x)\right)^2} - 1}$$

Written in the form of expectations, we have

$$\beta > \inf_{h(x)} \frac{\frac{\mathbb{E}_{x\sim p(x)}[h(x)^2]}{\left(\mathbb{E}_{x\sim p(x)}[h(x)]\right)^2} - 1}{\int \frac{dy}{p(y)}\left(\frac{\mathbb{E}_{x\sim p(x)}[p(y|x)h(x)]}{\mathbb{E}_{x\sim p(x)}[h(x)]}\right)^2 - 1}$$

$$= \inf_{h(x)} \frac{\frac{\mathbb{E}_{x\sim p(x)}[h(x)^2]}{\left(\mathbb{E}_{x\sim p(x)}[h(x)]\right)^2} - 1}{\mathbb{E}_{y\sim p(y)}\left[\left(\frac{\mathbb{E}_{x\sim p(x|y)}[h(x)]}{\mathbb{E}_{x\sim p(x)}[h(x)]}\right)^2\right] - 1}$$

We can absorb the constraint Eq. (10) into the above formula, and get

$$\beta > \inf_{h(x)} \beta_0[h(x)]$$

where

$$\beta_0[h(x)] = \frac{\frac{\mathbb{E}_{x\sim p(x)}[h(x)^2]}{\left(\mathbb{E}_{x\sim p(x)}[h(x)]\right)^2} - 1}{\mathbb{E}_{y\sim p(y)}\left[\left(\frac{\mathbb{E}_{x\sim p(x|y)}[h(x)]}{\mathbb{E}_{x\sim p(x)}[h(x)]}\right)^2\right] - 1}$$

which proves the condition of Theorem 5.

Furthermore, from Eq. (11) we have

$$\beta_0[h(x)] > 1$$

for $h(x) \not\equiv$ const, which satisfies the necessary condition of $\beta > 1$ in Section 2.

**Proof of lower bound of slope of the Pareto frontier at the origin:**

Now we prove the second statement of Theorem 5. Since $\delta I(X;Z) = 0$ and $\delta I(Y;Z) = 0$ according to Lemma 1.1, we have $\left( \frac{\Delta I(Y;Z)}{\Delta I(X;Z)} \right)^{-1} = \left( \frac{\delta^2 I(Y;Z)}{\delta^2 I(X;Z)} \right)^{-1}$. Substituting into the expression of $\delta^2 I(Y;Z)$ and $\delta^2 I(X;Z)$ from Lemma 4.1, we have

$$
\begin{aligned}
&\left( \frac{\Delta I(Y;Z)}{\Delta I(X;Z)} \right)^{-1} \\
&= \left( \frac{\delta^2 I(Y;Z)}{\delta^2 I(X;Z)} \right)^{-1} \\
&= \frac{\frac{\epsilon^2}{2} \int dx dz \frac{p(x)^2}{p(x)p(z)} h(z|x)^2 - \frac{\epsilon^2}{2} \int dx dx' dz \frac{p(x)p(x')}{p(z)} h(z|x)h(z|x')}{\frac{\epsilon^2}{2} \int dx dx' dy dz \frac{p(x,y)p(x',y)}{p(y)p(z)} h(z|x)h(z|x') - \frac{\epsilon^2}{2} \int dx dx' dz \frac{p(x)p(x')}{p(z)} h(z|x)h(z|x')} \\
&= \frac{\left( \int dx p(x)h(x)^2 - \int dx dx' p(x)p(x')h(x)h(x') \right) \int \frac{h_2(z)^2}{p(z)} dz}{\left( \int dx dx' dy \frac{p(x,y)p(x',y)}{p(y)} h(x)h(x') - \int dx dx' p(x)p(x')h(x)h(x') \right) \int \frac{h_2(z)^2}{p(z)} dz} \\
&= \frac{\int dx p(x)h(x)^2 - \int dx dx' p(x)p(x')h(x)h(x')}{\int dx dx' dy \frac{p(x,y)p(x',y)}{p(y)} h(x)h(x') - \int dx dx' p(x)p(x')h(x)h(x')} \\
&= \frac{\mathbb{E}_{x \sim p(x)}[h(x)^2] - \left( \mathbb{E}_{x \sim p(x)}[h(x)] \right)^2}{\mathbb{E}_{y \sim p(y)} \left[ \left( \mathbb{E}_{x \sim p(x|y)}[h(x)] \right)^2 \right] - \left( \mathbb{E}_{x \sim p(x)}[h(x)] \right)^2} \\
&= \frac{\frac{\mathbb{E}_{x \sim p(x)}[h(x)^2]}{\left( \mathbb{E}_{x \sim p(x)}[h(x)] \right)^2} - 1}{\mathbb{E}_{y \sim p(y)} \left[ \left( \frac{\mathbb{E}_{x \sim p(x|y)}[h(x)]}{\mathbb{E}_{x \sim p(x)}[h(x)]} \right)^2 \right] - 1} \\
&= \beta_0[h(x)]
\end{aligned}
$$

Therefore, $\left( \inf_{h(x)} \beta_0[h(x)] \right)^{-1}$ gives the largest slope of $\Delta I(Y;Z)$ vs. $\Delta I(X;Z)$ for perturbation function of the form $h_1(z|x) = h(x)h_2(z)$ satisfying $\int h_2(z)dz = 0$ and $\int \frac{h_2^2(z)}{p(z)} dz > 0$, which is a lower bound of slope of $\Delta I(Y;Z)$ vs. $\Delta I(X;Z)$ for all possible perturbation function $h_1(z|x)$. The latter is the slope of the Pareto frontier of the $I(Y;Z)$ vs. $I(X;Z)$ curve at the origin.

**Inflection point for general $Z$:** If we *do not* assume that $Z$ is at the origin of the information plane, but at some general stationary solution $Z^*$ with $p(z|x)$, we define

$$\beta^{(2)}[h(x)] = \left(\frac{\delta^2 I(Y;Z)}{\delta^2 I(X;Z)}\right)^{-1}$$

$$= \frac{\frac{\epsilon^2}{2}\int dxdz \frac{p(x)^2}{p(x,z)}h(z|x)^2 - \frac{\epsilon^2}{2}\int dxdx'dz\frac{p(x)p(x')}{p(z)}h(z|x)h(z|x')}{\frac{\epsilon^2}{2}\int dxdx'dydz\frac{p(x,y)p(x',y)}{p(y,z)}h(z|x)h(z|x') - \frac{\epsilon^2}{2}\int dxdx'dz\frac{p(x)p(x')}{p(z)}h(z|x)h(z|x')}$$

$$= \frac{\int dxdz\frac{p(x)^2}{p(x,z)}h(x)^2 - \int dxdx'dz\frac{p(x)p(x')}{p(z)}h(x)h(x')}{\int dxdx'dydz\frac{p(x,y)p(x',y)}{p(y,z)}h(x)h(x') - \int dxdx'dz\frac{p(x)p(x')}{p(z)}h(x)h(x')}$$

$$= \frac{\int \frac{dz}{p(z)}\left[\int dx\frac{p(x)^2}{p(x|z)}h(x)^2 - \left(\int dxp(x)h(x)\right)^2\right]}{\int \frac{dz}{p(z)}\left[\int \frac{dy}{p(y|z)}\left(\int dxp(x,y)h(x)\right)^2 - \left(\int dxp(x)h(x)\right)^2\right]}$$

$$= \frac{\int \frac{dz}{p(z)}\left[\frac{\int dx\frac{p(x)^2}{p(x|z)}h(x)^2}{\left(\int dxp(x)h(x)\right)^2} - 1\right]}{\int \frac{dz}{p(z)}\left[\frac{\int \frac{dy}{p(y|z)}\left(\int dxp(x,y)h(x)\right)^2}{\left(\int dxp(x)h(x)\right)^2} - 1\right]}$$

$$= \frac{\int dz\left[\frac{\int dx\frac{p(x)}{p(z|x)}h(x)^2}{\left(\int dxp(x)h(x)\right)^2} - \frac{1}{p(z)}\right]}{\int dz\left[\frac{\int \frac{dy}{p(z|y)p(y)}\left(\int dxp(x,y)h(x)\right)^2}{\left(\int dxp(x)h(x)\right)^2} - \frac{1}{p(z)}\right]}$$

$$= \frac{\int dz\left[\int dx\frac{p(x)}{p(z|x)}h(x)^2 - \frac{1}{p(z)}(\int dxp(x)h(x))^2\right]}{\int dz\left[\int \frac{dy}{p(z|y)p(y)}\left(\int dxp(x,y)h(x)\right)^2 - \frac{1}{p(z)}\left(\int dxp(x)h(x)\right)^2\right]}$$

which reduces to $\beta_0[h(x)]$ when $p(z|x) = p(z)$. When

$$\beta > \inf_{h(x)} \beta^{(2)}[h(x)] \tag{12}$$

It becomes a non-stable solution (non-minimum), and we will have other $Z$ that achieves a better $\text{IB}_\beta(X,Y;Z)$ than the current $Z^*$.

**Multiple phase transitions** To discuss multiple phase transitions, let us first obtain the $\beta^{(1)}$ for stationary solution for the IB objective. At a stationary solution for $\text{IB}_\beta[p(z|x)]$, for valid perturbation $h(z|x)$ satisfying $\int dzh(z|x) = 0$ for any $x$, we have $\delta\left[\text{IB}_\beta[p(z|x)] - \int dzdx\lambda(x)p(z|x)\right] = 0$ as a constraint optimization with $\lambda(x)$ as Lagrangian multipliers. Using Eq. (4), we have

$$\delta\text{IB}_\beta[p(z|x)] - \delta\int dzdx\lambda(x)p(z|x)$$

$$= \int dxdzp(x)h(z|x)\log\frac{p(z|x)}{p(z)} - \beta\int dxdydzp(x,y)h(z|x)\log\frac{p(z|y)}{p(z)} - \int dzdx\lambda(x)h(z|x) = 0$$

Therefore, we have

$$\beta^{(1)} \equiv \frac{\int dxdzp(x)h(z|x)\log\frac{p(z|x)}{p(z)} - \int dzdx\lambda(x)h(z|x)}{\int dxdydzp(x,y)h(z|x)\log\frac{p(z|y)}{p(z)}}$$

$$= \frac{p(x)\log\frac{p(z|x)}{p(z)} - \lambda(x)}{\int dyp(x,y)\log\frac{p(z|y)}{p(z)}} \tag{13}$$

The last equality is due to that the first equality is always true for any function $h(z|x)$. So we can take out the $\int dxdzh(z|x)$ factor. $\lambda(x)$ is used for normalization of $p(z|x)$. Eq. (13) is equivalent to the result of the self-consistent equation in Tishby et al. (2000).

Eq. (13) and Eq. (12) provide us with an ideal tool to study multiple phase transitions. For each $\beta$, at the minimization of the IB objective, Eq. (13) is satisfied by some $Z^*$ that is at the Pareto frontier on the $I(Y; Z)$ vs. $I(X; Z)$ plane. As we increase $\beta$, the $\inf_{h(x)} \beta^{(2)}[h(x)]$ may remain stable for a wide range of $\beta$, until $\beta$ is greater than $\inf_{h(x)} \beta^{(2)}[h(x)]$, at which point we will have a phase transition where suddenly there is a better $Z = Z^{**}$ that achieves much lower $\text{IB}_\beta(X, Y; Z)$ value.

For example, we can rewrite Eq. (13) as

$$\log\frac{p(z|x)}{p(z)} = \beta^{(1)} \int dy p(y|x) \log\frac{p(z|y)}{p(z)} + \tilde{\lambda}(x) \tag{14}$$

where $\tilde{\lambda}(x) = \frac{\lambda(x)}{p(x)}$. By substituting into Eq. (12), we may proceed and get useful results.

$\square$

## G  PROOF OF THEOREM 2 (CONFIDENT SUBSET SUFFICIENT CONDITION)

*Proof.* According to Theorem 5, a sufficient condition for $(X, Y)$ to be $\text{IB}_\beta$-learnable is that $X$ and $Y$ are not independent, and

$$\beta > \inf_{h(x)} \frac{\frac{\mathbb{E}_{x\sim p(x)}[h(x)^2]}{\left(\mathbb{E}_{x\sim p(x)}[h(x)]\right)^2} - 1}{\mathbb{E}_{y\sim p(y)}\left[\left(\frac{\mathbb{E}_{x\sim p(x|y)}[h(x)]}{\mathbb{E}_{x\sim p(x)}[h(x)]}\right)^2\right] - 1} \tag{15}$$

We can assume a specific form of $h(x)$, and obtain a (potentially stronger) sufficient condition. Specifically, we let

$$h(x) = \begin{cases} 1, x \in \Omega_x \\ 0, \text{otherwise} \end{cases} \tag{16}$$

for certain $\Omega_x \subset \mathcal{X}$. Substituting into Eq. (16), we have that a sufficient condition for $(X, Y)$ to be $\text{IB}_\beta$-learnable is

$$\beta > \inf_{\Omega_x \subset \mathcal{X}} \frac{\frac{p(\Omega_x)}{p(\Omega_x)^2} - 1}{\int dy p(y) \left(\frac{\int_{x\in\Omega_x} dx p(x|y) dx}{p(\Omega_x)}\right)^2 - 1} > 0 \tag{17}$$

where $p(\Omega_x) = \int_{x\in\Omega_x} p(x) dx$.

The denominator of Eq. (17) is

$$\int dy p(y) \left(\frac{\int_{x\in\Omega_x} dx p(x|y) dx}{p(\Omega_x)}\right)^2 - 1$$

$$= \int dy p(y) \left(\frac{p(\Omega_x|y)}{p(\Omega_x)}\right)^2 - 1$$

$$= \int dy \frac{p(y|\Omega_x)^2}{p(y)} - 1$$

$$= \mathbb{E}_{y\sim p(y|\Omega_x)}\left[\frac{p(y|\Omega_x)}{p(y)} - 1\right]$$

Using the inequality $x - 1 \geq \log x$, we have

$$\mathbb{E}_{y\sim p(y|\Omega_x)}\left[\frac{p(y|\Omega_x)}{p(y)} - 1\right] \geq \mathbb{E}_{y\sim p(y|\Omega_x)}\left[\log\frac{p(y|\Omega_x)}{p(y)}\right] \geq 0$$

Both equalities hold iff $p(y|\Omega_x) \equiv p(y)$, at which the denominator of Eq. (17) is equal to 0 and the expression inside the infimum diverge, which will not contribute to the infimum. Except this scenario, the denominator is greater than 0. Substituting into Eq. (17), we have that a sufficient condition for $(X, Y)$ to be $\mathrm{IB}_\beta$-learnable is

$$\beta > \inf_{\Omega_x \subset \mathcal{X}} \frac{\frac{p(\Omega_x)}{p(\Omega_x)^2} - 1}{\mathbb{E}_{y \sim p(y|\Omega_x)} \left[ \frac{p(y|\Omega_x)}{p(y)} - 1 \right]} \tag{18}$$

Since $\Omega_x$ is a subset of $\mathcal{X}$, by the definition of $h(x)$ in Eq. (16), $h(x)$ is not a constant in the entire $\mathcal{X}$. Hence the numerator of Eq. (18) is positive. Since its denominator is also positive, we can then neglect the "$> 0$", and obtain the condition in Theorem 2.

Since the $h(x)$ used in this theorem is a subset of the $h(x)$ used in Theorem 5, the infimum for Eq. (2) is greater than or equal to the infimum in Eq. (5). Therefore, according to the second statement of Theorem 5, we have that the $(\inf_{\Omega_x} \beta_0(\Omega_x))^{-1}$ is also a lower bound of the slope for the Pareto frontier of $I(Y; Z)$ vs. $I(X; Z)$ curve.

Now we prove that the condition Eq. (2) is invariant to invertible mappings of $X$. In fact, if $X' = g(X)$ is a uniquely invertible map (if $X$ is continuous, $g$ is additionally required to be continuous), let $\mathcal{X}' = \{g(x)|x \in \Omega_x\}$, and denote $g(\Omega_x) \equiv \{g(x)|x \in \Omega_x\}$ for any $\Omega_x \subset \mathcal{X}$, we have $p(g(\Omega_x)) = p(\Omega_x)$, and $p(y|g(\Omega_x)) = p(y|\Omega_x)$. Then for dataset $(X, Y)$, let $\Omega'_x = g(\Omega_x)$, we have

$$\frac{\frac{1}{p(\Omega'_x)} - 1}{\mathbb{E}_{y \sim p(y|\Omega'_x)} \left[ \frac{p(y|\Omega'_x)}{p(y)} - 1 \right]} = \frac{\frac{1}{p(\Omega_x)} - 1}{\mathbb{E}_{y \sim p(y|\Omega_x)} \left[ \frac{p(y|\Omega_x)}{p(y)} - 1 \right]} \tag{19}$$

Additionally we have $\mathcal{X}' = g(\mathcal{X})$. Then

$$\inf_{\Omega'_x \subset \mathcal{X}'} \frac{\frac{1}{p(\Omega'_x)} - 1}{\mathbb{E}_{y \sim p(y|\Omega'_x)} \left[ \frac{p(y|\Omega'_x)}{p(y)} - 1 \right]} = \inf_{\Omega_x \subset \mathcal{X}} \frac{\frac{1}{p(\Omega_x)} - 1}{\mathbb{E}_{y \sim p(y|\Omega_x)} \left[ \frac{p(y|\Omega_x)}{p(y)} - 1 \right]} \tag{20}$$

For dataset $(X', Y) = (g(X), Y)$, applying Theorem 2 we have that a sufficient condition for it to be $\mathrm{IB}_\beta$-learnable is

$$\beta > \inf_{\Omega'_x \subset \mathcal{X}'} \frac{\frac{1}{p(\Omega'_x)} - 1}{\mathbb{E}_{y \sim p(y|\Omega'_x)} \left[ \frac{p(y|\Omega'_x)}{p(y)} - 1 \right]} = \inf_{\Omega_x \subset \mathcal{X}} \frac{\frac{1}{p(\Omega_x)} - 1}{\mathbb{E}_{y \sim p(y|\Omega_x)} \left[ \frac{p(y|\Omega_x)}{p(y)} - 1 \right]} \tag{21}$$

where the equality is due to Eq. (20). Comparing with the condition for $\mathrm{IB}_\beta$-learnability for $(X, Y)$ (Eq. (2)), we see that they are the same. Therefore, the condition given by Theorem 2 is invariant to invertible mapping of $X$.

## H    PROOF OF COROLLARY 2.1 AND COROLLARY 2.2

### H.1    PROOF OF COROLLARY 2.1

*Proof.* We use Theorem 2. Let $\Omega_x$ contain all elements $x$ whose true class is $y^*$ for some certain $y^*$, and 0 otherwise. Then we obtain a (potentially stronger) sufficient condition. Since the probability $p(y|y^*, x) = p(y|y^*)$ is class-conditional, we have

$$\inf_{\Omega_x \subset \mathcal{X}} \frac{\frac{1}{p(\Omega_x)} - 1}{\mathbb{E}_{y \sim p(y|\Omega_x)} \left[ \frac{p(y|\Omega_x)}{p(y)} - 1 \right]}$$

$$= \inf_{y^*} \frac{\frac{1}{p(y^*)} - 1}{\mathbb{E}_{y \sim p(y|y^*)} \left[ \frac{p(y|y^*)}{p(y)} - 1 \right]}$$

By requiring $\beta > \inf_{y^*} \frac{\frac{1}{p(y^*)} - 1}{\mathbb{E}_{y \sim p(y|y^*)}\left[\frac{p(y|y^*)}{p(y)} - 1\right]}$, we obtain a sufficient condition for IB$_\beta$ learnability. $\qquad\square$

## H.2 Proof of Corollary 2.2

*Proof.* We again use Theorem 2. Since $Y$ is a deterministic function of $X$, let $Y = f(X)$. Since it is classification problem, $Y$ contains at least one value $y$ such that its probability $p(y) > 0$, we let $\Omega_x$ contain only $x$ such that $f(x) = y$. Substituting into Eq. (2), we have

$$\frac{\frac{1}{p(\Omega_x)} - 1}{\mathbb{E}_{y \sim p(y|\Omega_x)}\left[\frac{p(y|\Omega_x)}{p(y)} - 1\right]}$$

$$= \frac{\frac{1}{p(y)} - 1}{\mathbb{E}_{y \sim p(y|\Omega_x)}\left[\frac{1}{p(y)} - 1\right]}$$

$$= \frac{\frac{1}{p(y)} - 1}{\frac{1}{p(y)} - 1}$$

$$= 1$$

$\qquad\square$

Therefore, the sufficient condition becomes $\beta > 1$.

Furthermore, since a necessary condition for IB$_\beta$-learnability is $\beta > 1$ (Section 2), we have that $\beta > \beta_0 = 1$ is a necessary and sufficient condition.

$\qquad\square$

## I  Additional discussions for Theorem 2

**Similarity to information measures.**  The denominator of Eq. (2) is closely related to mutual information. Using the inequality $x - 1 \geq \log(x)$ for $x > 0$, it becomes:

$$\mathbb{E}_{y \sim p(y|\Omega_x)}\left[\frac{p(y|\Omega_x)}{p(y)} - 1\right] \geq \mathbb{E}_{y \sim p(y|\Omega_x)}\left[\log \frac{p(y|\Omega_x)}{p(y)}\right]$$
$$= \tilde{I}(\Omega_x; Y)$$

where $\tilde{I}(\Omega_x; Y)$ is the mutual information "density" at $\Omega_x \subset \mathcal{X}$. Of course, this quantity is also $\mathbb{D}_{\mathrm{KL}}[p(y|\Omega_x)||p(y)]$, so we know that the denominator of Eq. (2) is non-negative. Incidentally, $\mathbb{E}_{y \sim p(y|\Omega_x)}\left[\frac{p(y|\Omega_x)}{p(y)} - 1\right]$ is the density of "rational mutual information" (Lin & Tegmark (2016)) at $\Omega_x$.

Similarly, the numerator is related to the self-information of $\Omega_x$:

$$\frac{1}{p(\Omega_x)} - 1 \geq \log \frac{1}{p(\Omega_x)} = -\log p(\Omega_x) = h(\Omega_x)$$

so we can estimate the phase transition as:

$$\beta \gtrapprox \inf_{\Omega_x \subset \mathcal{X}} \frac{h(\Omega_x)}{\tilde{I}(\Omega_x; Y)} \tag{22}$$

Since Eq. (22) uses upper bounds on both the numerator and the denominator, it does not give us a bound on $\beta_0$.

**Multiple phase transitions.**  Based on this characterization of $\Omega_x$, we can hypothesize datasets with multiple learnability phase transitions. Specifically, consider a region $\Omega_{x0}$ that is small but "typical", consists of all elements confidently predicted as $y_0$ by $p(y|x)$, and where $y_0$ is the least common class. By construction, this $\Omega_{x0}$ will dominate the infimum in Eq. (2), resulting in a small value of $\beta_0$. However, the remaining $\mathcal{X} - \Omega_{x0}$ effectively form a new dataset,

$\mathcal{X}_1$. At exactly $\beta_0$, we may have that the current encoder, $p_0(z|x)$, has no mutual information with the remaining classes in $\mathcal{X}_1$; i.e., $I(Y_1; Z_0) = 0$. In this case, Definition 1 applies to $p_0(z|x)$ with respect to $I(X_1; Z_1)$. We might expect to see that, at $\beta_0$, learning will plateau until we get to some $\beta_1 > \beta_0$ that defines the phase transition for $\mathcal{X}_1$. Clearly this process could repeat many times, with each new dataset $\mathcal{X}_i$ being distinctly more difficult to learn than $\mathcal{X}_{i-1}$. The end of Appendix F gives a more detailed analysis on multiple phase transitions.

**Estimating model capacity.** The observation that a model can't distinguish between cluster overlap in the data and its own lack of capacity gives an interesting way to use IB-Learnability to measure the capacity of a set of models relative to the task they are being used to solve.

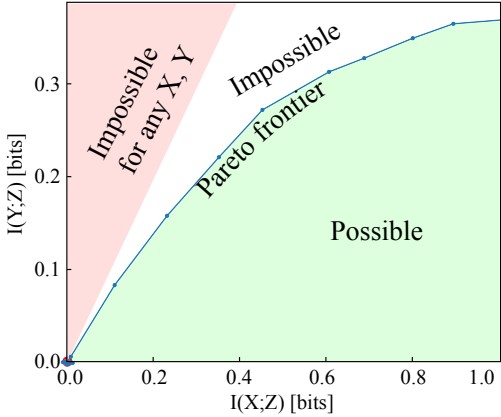

Figure 2: The Pareto frontier of mutual information that $Z$ can have with $X$ and $Y$ is shown for the MNIST example from Figure 1. $H(Y) = 1$ bit since only the two of ten digits are used, and $I(Y, Z) \leq I(Y; X) \approx 0.5$ bits $< H(Y)$ because of the 20% label swapping. The true frontier is differentiable; the figure shows a variational approximation that places an upper bound on both informations, horizontally offset to pass through the origin.

**Learnability and the Information Plane.** Many of our results can be interpreted in terms of the geometry of the Pareto frontier illustrated in Fig. 2, which describes the trade-off between increasing $I(Y; Z)$ and decreasing $I(X; Z)$. At any point on this frontier that minimizes $\mathrm{IB}_\beta^{\min} \equiv \min I(X; Z) - \beta I(Y; Z)$, the frontier will have slope $\beta^{-1}$ if it is differentiable. If the frontier is also concave (has negative second derivative), then this slope $\beta^{-1}$ will take its maximum $\beta_0^{-1}$ at the origin, which implies $\mathrm{IB}_\beta$-Learnability for $\beta > \beta_0$, so that the threshold for $\mathrm{IB}_\beta$-Learnability is simply the inverse slope of the frontier at the origin. More generally, as long as the Pareto frontier is differentiable, the threshold for $\mathrm{IB}_\beta$-learnability is the inverse of its maximum slope. Indeed, Theorem 2 gives lower bounds of the slope of the Pareto frontier at the origin.

The shaded region $I(Y; Z) > I(X; Z)$ is impossible by the data processing inequality, since $Z$ depends on $Y$ only via $X$. Indeed, it is easy to see that if the Pareto Frontier is differentiable, its slope must always lie in the unit interval $[0, 1]$. Moreover, both the horizontal and vertical axes are bounded, since $0 \leq I(X; Z) \leq H(X)$ and $0 \leq I(Y; Z) \leq H(Y)$, so if the Pareto Frontier is differentiable, then its slope lies in some compact interval $[\beta_1^{-1}, \beta_0^{-1}]$, where $1 \leq \beta_0 \leq \beta_1$. This means that we lack $\mathrm{IB}_\beta$-learnability for $\beta < \beta_0$, which makes the origin the optimal point. If the frontier is convex, then we achieve optimality at the upper right endpoint if $\beta > \beta_1$, otherwise on the frontier at the location between the two endpoints where the frontier slope is $\beta^{-1}$.

**Learnability and contraction coefficient** If we regard the true mapping from $X$ to $Y$ as a channel with transition kernel $P_{Y|X}$, we can define contraction coefficient $\eta_{\mathrm{KL}}(P_{Y|X}) = \sup_{Q; P: 0 < \mathbb{D}_{\mathrm{KL}}(P||Q) < \infty} \frac{\mathbb{D}_{\mathrm{KL}}(P_{Y|X} \circ P || P_{Y|X} \circ Q)}{\mathbb{D}_{\mathrm{KL}}(P||Q)}$ (Polyanskiy & Wu (2017)) as a measure of how much it keeps the two distributions $P$ and $Q$ intact (as opposed to being drawn nearer measured by KL-divergence) after pushing forward through the channel. By Polyanskiy & Wu (2017) we have $\eta_{\mathrm{KL}}(P_{Y|X}) = \sup_Z \frac{I(Y; Z)}{I(X; Z)}$, which is the slope $\beta_0^{-1}$ of the Pareto frontier at the origin. By the analysis of the information plane above, we have that as long as $\beta^{-1} < \eta_{\mathrm{KL}}(P_{Y|X}) = \beta_0^{-1}$, it is $\mathrm{IB}_\beta$-learnable. Furthermore, with Theorem 5, we have $\left(\inf_{h(x)} \beta_0[h(x)]\right)^{-1} \leq \sup_{h(z|x)} \frac{\Delta I(Y; Z)}{\Delta I(X; Z)}\big|_{p(z|x) = p(z)} \leq \sup_Z \frac{I(Y; Z)}{I(X; Z)}$, therefore

$\left(\inf_{h(x)} \beta_0[h(x)]\right)^{-1} \leq \eta_{\mathrm{KL}}(P_{Y|X})$. Theorem 5 hence also provides a lower bound for the contraction coefficient $\eta_{\mathrm{KL}}(P_{Y|X})$. Similarly for Theorem 2.

## J   ALGORITHM FOR ESTIMATING THE IB-LEARNABILITY CONDITION

In Alg. 1 we present a detailed algorithm for estimating $\beta_0$.

---

**Algorithm 1 Estimating the upper bound for $\beta_0$ for IB$_\beta$-Learnability**

---

   **Require**: Dataset $\mathcal{D} = \{(x_i, y_i)\}, i = 1, 2, ...N$. The number of classes is $C$.
   **Require** $\varepsilon$: tolerance for estimating $\beta_0$
  1: Learn a maximum likelihood model $p_\theta(y|x)$ using the dataset $\mathcal{D}$.
  2: Construct matrix $(P_{y|x})$ such that $(P_{y|x})_{ij} = p_\theta(y = j|x = x_i)$.
  3: Construct vector $p_y = (p_{y1}, .., p_{yC})$ such that $p_{yj} = \frac{1}{N}\sum_{i=1}^{N}(P_{y|x})_{ij}$.
  4: $i^* = \arg\max_i \mathbf{Get\beta}(P_{y|x}, p_y, \{i\})$.
  5: $j^* = \arg\max_j (P_{y|x})_{ij}$
  6: Sort the rows of $P_{y|x}$ in decreasing values of $(P_{y|x})_{ij^*}$.
  7: Search $i_{\mathrm{upper}}$ until $\tilde{\beta}_0 = \mathbf{Get\beta}(P_{y|x}, p_y, \Omega)$ is minimal with tolerance $\varepsilon$, where $\Omega = \{1, 2, ...i_{\mathrm{upper}}\}$.
  8: **return** $\tilde{\beta}_0$

   **Subroutine Get$\beta(P_{y|x}, p_y, \Omega)$**
  s1: $(N, C, n) \leftarrow$ (number of rows of $P_{y|x}$, number of columns of $P_{y|x}$, number of elements of $\Omega$).
  s2: $(p_{y|\Omega})_j \leftarrow \frac{1}{n}\sum_{i \in \Omega}(P_{y|x})_{ij}, j = 1, 2, ..., C$.
  s3: $\tilde{\beta}_0 \leftarrow \dfrac{\frac{N}{n} - 1}{\sum_j \left[\frac{(p_{y|\Omega_x})_j^2}{p_{yj}} - 1\right]}$
  s4: **return** $\tilde{\beta}_0$

---

## K   SETTINGS FOR THE EXPERIMENTS

We use the Variational Information Bottleneck (VIB) objective by Alemi et al. (2016). For the synthetic experiment, the latent $Z$ has dimension of 2. The encoder is a neural net with 2 hidden layers, each of which has 128 neurons with ReLU activation. The last layer has linear activation and 4 output neurons, with the first two parameterizes the mean of a Gaussian and the last two parameterizes the log variance of the Gaussian. The decoder is a neural net with 1 hidden layers with 128 neurons and ReLU activation. Its last layer has linear activation and outputs the logit for the class labels. It uses a mixture of Gaussian prior with 500 components (for the experiment with class overlap, 256 components), each of which is a 2D Gaussian with learnable mean and log variance, and the weights for the components are also learnable. For the MNIST experiment, the architecture is mostly the same, except the following: (1) for $Z$, we let it have dimension of 256. For the prior, we use standard Gaussian with diagonal covariance matrix.

For all experiments, we use Adam (Kingma & Ba (2014)) optimizer with default parameters. We do not add any regularization. We use learning rate of $10^{-4}$ and have a learning rate decay of $\frac{1}{1+0.01\times\mathrm{epoch}}$. We train in total 2000 epochs with batch size of 500. All experiments has train-test split of 5:1, and we report the accuracy on the test set, w.r.t. the true labels.

For estimation of $\beta_{0,\mathrm{exp}}$ in Fig. 1, in the accuracy vs. $\beta_i$ curve, we take the mean and standard deviation of the accuracy for the lowest 5 $\beta_i$ values, denoting as $\mu_\beta, \sigma_\beta$. When $\beta_i$ is greater than $\mu_\beta + 3\sigma_\beta$, we regard it as learning a non-trivial representation, and take the average of $\beta_i$ and $\beta_{i-1}$ as the experimentally estimated onset of learning. We also inspect manually and confirm that it is consistent with human intuition.

For the estimating $\beta_{0,\mathrm{estimated}}$ using Alg. 1, at step 7 we use the following discrete search algorithm. We gradually narrow down the range $[a, b]$ of $i_{\mathrm{upper}}$, starting from $[1, N]$. At each iteration, we set a tentative new range $[a', b']$, where $a' = 0.8a + 0.2b, b' = 0.2a + 0.8b$, and calculate $\tilde{\beta}_{0,a'} = \mathbf{Get\beta}(P_{y|x}, p_y, \Omega_{a'}), \tilde{\beta}_{0,b'} = \mathbf{Get\beta}(P_{y|x}, p_y, \Omega_{b'})$ where $\Omega_{a'} = \{1, 2, ...a'\}$ and $\Omega_{b'} = \{1, 2, ...b'\}$. If $\tilde{\beta}_{0,a'} < \tilde{\beta}_{0,a}$, let $a \leftarrow a'$. If $\tilde{\beta}_{0,b'} < \tilde{\beta}_{0,b}$, let $b \leftarrow b'$. In other words, we narrow down the range of $i_{\mathrm{upper}}$ if we find that the $\Omega$ given by the left or right boundary gives a lower $\tilde{\beta}_0$

Table 1: Class confusion matrix used in CIFAR10 experiments. The value in row $i$, column $j$ means for class $i$, the probability of mislabeling it as class $j$. The mean confusion across the classes is 20%.

|  | Plane | Auto. | Bird | Cat | Deer | Dog | Frog | Horse | Ship | Truck |
|---|---|---|---|---|---|---|---|---|---|---|
| Plane | 0.82232 | 0.00238 | 0.021 | 0.00069 | 0.00108 | 0 | 0.00017 | 0.00019 | 0.1473 | 0.00489 |
| Auto. | 0.00233 | 0.83419 | 0.00009 | 0.00011 | 0 | 0.00001 | 0.00002 | 0 | 0.00946 | 0.15379 |
| Bird | 0.03139 | 0.00026 | 0.76082 | 0.0095 | 0.07764 | 0.01389 | 0.1031 | 0.00309 | 0.00031 | 0 |
| Cat | 0.00096 | 0.0001 | 0.00273 | 0.69325 | 0.00557 | 0.28067 | 0.01471 | 0.00191 | 0.00002 | 0.0001 |
| Deer | 0.00199 | 0 | 0.03866 | 0.00542 | 0.83435 | 0.01273 | 0.02567 | 0.08066 | 0.00052 | 0.00001 |
| Dog | 0 | 0.00004 | 0.00391 | 0.2498 | 0.00531 | 0.73191 | 0.00477 | 0.00423 | 0.00001 | 0 |
| Frog | 0.00067 | 0.00008 | 0.06303 | 0.05025 | 0.0337 | 0.00842 | 0.8433 | 0 | 0.00054 | 0 |
| Horse | 0.00157 | 0.00006 | 0.00649 | 0.00295 | 0.13058 | 0.02287 | 0 | 0.83328 | 0.00023 | 0.00196 |
| Ship | 0.1288 | 0.01668 | 0.00029 | 0.00002 | 0.00164 | 0.00006 | 0.00027 | 0.00017 | 0.83385 | 0.01822 |
| Truck | 0.01007 | 0.15107 | 0 | 0.00015 | 0.00001 | 0.00001 | 0 | 0.00048 | 0.02549 | 0.81273 |

value. The process stops when both $\tilde{\beta}_{0,a'}$ and $\tilde{\beta}_{0,b'}$ stops improving (which we find always happens when $b' = a' + 1$), and we return the smaller of the final $\tilde{\beta}_{0,a'}$ and $\tilde{\beta}_{0,b'}$ as $\tilde{\beta}_0$.

### K.1 CIFAR10 DETAILS

We trained a deterministic 28x10 wide resnet (He et al., 2016; Zagoruyko & Komodakis, 2016), using the open source implementation from Cubuk et al. (2018). However, we extended the final 10 dimensional logits of that model through another 3 layer MLP classifier, in order to keep the inference network architecture identical between this model and the VIB models we describe below. During training, we dynamically added label noise according to the class confusion matrix in Tab. K.1. The mean label noise averaged across the 10 classes is 20%. After that model had converged, we used it to estimate $\beta_0$ with Alg. 1. Even with 20% label noise, $\beta_0$ was estimated to be 1.0483.

We then trained 73 different VIB models using the same 28x10 wide resnet architecture for the encoder, parameterizing the mean of a 10-dimensional unit variance Gaussian. Samples from the encoder distribution were fed to the same 3 layer MLP classifier architecture used in the deterministic model. The marginal distributions were mixtures of 500 fully covariate 10-dimensional Gaussians, all parameters of which are trained. The VIB models had $\beta$ ranging from 1.02 to 2.0 by steps of 0.02, plus an extra set ranging from 1.04 to 1.06 by steps of 0.001 to ensure we captured the empirical $\beta_0$ with high precision.

However, this particular VIB architecture does not start learning until $\beta > 2.5$, so none of these models would train as described.[4] Instead, we started them all at $\beta = 100$, and annealed $\beta$ down to the corresponding target over 10,000 training gradient steps. The models continued to train for another 200,000 gradient steps after that. In all cases, the models converged to essentially their final accuracy within 20,000 additional gradient steps after annealing was completed. They were stable over the remaining $\sim 180,000$ gradient steps.

---

[4]A given architecture trained using maximum likelihood and with no stochastic layers will tend to have higher effective capacity than the same architecture with a stochastic layer that has a fixed but non-trivial variance, even though those two architectures have exactly the same number of learnable parameters.

