# OpenReview forum: "Learnability for the Information Bottleneck"
_ICLR.cc/2019/Workshop/LLD — LLD 2019_

### Official Review · AnonReviewer1 · 2019-04-11
**Interesting analysis of learnability in IB with surprising conclusions**

**Rating:** 4
**Confidence:** 2

**Review:**

The paper analyses learnability of the IB regulariser. It is well-written and the results are interesting because the non-monotonic behavior is not something one would intuitively assume to appear. I definitely recommend acceptance of the paper, even though it may be slightly out of scope for the workshop due to its lack of experiments on limited data.

---

### Official Review · AnonReviewer2 · 2019-04-12
**Learnability for the Information Bottleneck**

**Rating:** 4
**Confidence:** 2

**Review:**

The information-bottleneck (IB) framework of Tishby and coworkers proposes to learn a compression / representation of the data (x, y) which captures as little information as possible about the input x but as much information as possible about the prediction target y. To this end, it proposes to minimize a functional of the form

min I(X; Z) - beta * I(Y; Z), ... (IB)

w.r.t to the conditional distribution P(z|x). It is known that the choice of the balancing parameter beta > 0 is crucial. In particular, if beta <= 1, then the global optimum of the above problem is an uninformative distribution P(z|x) = P(z), which discards the input x.

The current paper pushes the analysis further by proposing a new notion of learnability called IB-learnability: a problem is IB learnable at rank beta iff the stationary point* P(z|x) = P(z) is not a global optimum of problem (IB) above. The authors then go on to produce sufficient conditions under which a problem is learnable. Also, efficient heuristics are proposed for computing / checking these sufficient learnability conditions.

Though I didn't check the proofs (> 16 pages), the paper seems well-grounded from a formal perspective. Also, a rich array of experiments on both synthetic and real data (MNIST, CIFAR10, etc.) are presented and discussed.

My only worry is that the paper might be slightly out of the scope of what the LLD workshop is concerned with. That notwithstanding, I think the paper should be accepted and discussed at the workshop.

*A side result obtained by the paper is that for any beta, P(z|x) = p(z) is a stationary point for problem (IB).

---

### Decision · Program_Chairs · 2019-04-12
**Acceptance Decision**

Accept